

# Modelling the relationship between liquid water content and cloud droplet number concentration observed in low clouds in the summer Arctic and its radiative effects

Joelle Dionne[1], Knut von Salzen[2,3,4], Jason Cole[2], Rashed Mahmood[3,*], W. Richard Leaitch[2], Glen Lesins[1], Ian Folkins[1], Rachel Y.-W. Chang[1]

1- Physics and Atmospheric Science Department, Dalhousie University, Halifax, Canada

2- Climate Research Division, Science and Technology Branch, Environment and Climate Change Canada, Toronto, Canada

3- School of Earth and Ocean Sciences, University of Victoria, Victoria, Canada

4- Earth, Ocean, and Atmospheric Sciences Department, University of British Columbia, Vancouver, Canada

*- Now at Department of Atmospheric Science, School of Environmental Studies, China University of Geosciences, Wuhan, China

Correspondence to: R.Y.-W. Chang (rachel.chang@dal.ca)

**Abstract.** Low clouds persist in the summer Arctic with important consequences for the radiation budget. In this study, we simulate the linear relationship between liquid water content (LWC) and cloud droplet number concentration (CDNC) observed during an aircraft campaign based out of

Resolute Bay, Canada conducted as part of the NETCARE study in July 2014. Using a single column model, we find that autoconversion can explain the observed linear relationship between LWC and CDNC. Of the three schemes we examined, the autoconversion scheme using continuous drizzle (Khairoutdinov and Kogan, 2000) appears to best reproduce the observed linearity in the tenuous-cloud regime (Mauritsen et al., 2011), while a scheme with a threshold for rain (Liu and Daum, 2004) best

reproduces the linearity at higher CDNC. An offline version of the radiative transfer model used in the Canadian Atmospheric Model version 4.3 is used to compare the radiative effects of the modelled and observed clouds,.We find that there is no significant difference in the upward longwave fluxes at the top of the atmosphere from the three autoconversion schemes ($p=0.05$), but that all three schemes differ at $p=0.05$ from the calculations based on observations. In contrast, the downward longwave and

shortwave fluxes at the surface for all three schemes do not differ significantly ($p=0.01$) from the observation-based radiative calculations.



## 1 Introduction

Observations show a warming trend in the Arctic that is 2.5 times greater than the rest of the world
(ACIA, 2005). One known uncertainty in our understanding of climate change is the effect of clouds on

the radiation budget (Lohmann and Hoose, 2009), with particularly important consequences for Arctic
climate. Microphysical properties of Arctic clouds are sensitive to changes in cloud condensation
nuclei (CCN) concentrations (Coopman et al., 2018). Like at other latitudes, smaller cloud droplets in
the Arctic are associated with less shortwave radiation at the surface than larger droplets due to an
increased reflectivity (Peng et al., 2002). However, the net radiative effect of cloud droplet size and

number concentration can vary in sign in the Arctic when combining the longwave and shortwave
radiative effects because of the high surface albedo and solar zenith angle (Curry et al., 1996). Overall,
the radiative forcing from shortwave radiation due to cloud is dominated by cloud microphysical
properties such as liquid water content (LWC), effective radius, cloud droplet number concentration
(CDNC), as well as solar zenith angle, and surface albedo (Curry et al., 1996). Similarly, the longwave

cloud radiative forcing is dominated by LWC, effective radius, phase, and emission temperature of the
cloud (Sedlar et al., 2010). In addition, model runs without shortwave radiation have shown that it can
also impact Arctic stratus clouds by limiting their height as well as microphysical properties,
demonstrating feedbacks between radiation and cloud properties (Olsson et al., 1998). In general, the
impact of increasing the CDNC is more complicated than just reducing the cloud droplet size and

increasing the cloud reflectance, as it may inhibit precipitation, cause smaller droplets to evaporate
faster in non-precipitating clouds, and/or suppress the breakup of clouds by precipitation (Rosenfeld et
al., 2014).

In cloud models, an important parameterization that affects the cloud microphysical properties, and

thus cloud lifetime and radiative effects, is the autoconversion scheme, which converts cloud droplets
to drizzle drops in order to simulate rain. These schemes are usually used instead of explicit
calculations of the cloud droplet size distribution to reduce the computational cost and complexity of
models. Autoconversion schemes can depend on variables such as cloud LWC, air density, CDNC, and
droplet radius. Some have a threshold below which the cloud does not simulate rain while others

simulate continuous precipitation based on LWC. Autoconversion rates from different
parameterizations can vary from $10^{-7}$ to $10^{-11}$ kg m$^{-3}$ s$^{-1}$ for marine boundary layer clouds (Wood,
2005b), so the choice of autoconversion scheme can be significant. A recent study compared the output
of six models simulating clean Arctic conditions, showing that under very clean conditions, clouds can



be very sensitive to cloud condensation nuclei (CCN) concentrations, with otherwise-identical

simulations from individual models producing different cloud properties, to the point that the LWC and

radiative effects of the clouds were CCN-limited (Stevens et al., 2018). In that study, models with

faster autoconversion rates were found to be generally less sensitive to changes in CDNC or CCN

concentrations for all examined cloud properties. However, the model simulations did not allow

different autoconversion parameterizations to be compared using the same model. Furthermore, that

study did not consider the compare the results of Arctic clouds with different CCN concentrations or

rain formation schemes in the models (Stevens et al., 2018).

Recent observations by Leaitch et al. (2016) showed a strong linear relationship between LWC and

CDNC in low altitude liquid clouds in the summertime Canadian Arctic. Instead of droplet size

reducing with increasing CDNC, the volume mean diameter remained approximately constant, with a

value near 20 µm (Leaitch et al., 2016). Three possible physical explanations for the linear relationship

between LWC and CDNC are discussed here. One possible cause is autoconversion, since the

autoconversion of cloud water becomes less efficient at higher CDNC because relatively fewer droplets

are converted to rain drops, so the liquid in them stays as LWC rather than precipitating out, leading to

higher cloud LWC (Albrecht, 1989). A second possible cause is the entrainment of dry air parcels into

a cloud without mixing with the cloud droplets. This type of inhomogeneous mixing occurs when the

evaporation timescale is shorter than the timescale to mix the entrained parcels within the cloud, which

results in some droplets evaporating fully in and near the entrained parcel, lowering the CDNC as well

as the LWC (Gerber et al., 2008; Jensen et al., 1985), which may lead to a nearly linear relationship

between LWC and CDNC. In contrast, during homogeneous mixing, the evaporation timescale is

longer than the mixing timescale, which results in most cloud droplets losing some water, but not

completely evaporating, thus lowering the LWC while keeping the CDNC constant. During one of the

flights, Leaitch et al. (2016) noted that entrainment appeared to reduce the CDNC, but not the LWC,

which is inconsistent with the linear change observed overall. As such, while entrainment may be a

possible driver of the linearity of the LWC-CDNC relationship on the other days, it is likely not the

sole or main driver overall in our dataset. A final possible cause is increased rates of cooling causing

increased rates of condensation (and possibly supersaturation), which increases both the CDNC and

LWC. A possible mechanism for this would be fog advecting over a colder surface, as when a water

temperature gradient exists. The implication of autoconversion driving part of the observed linear



relationship is that it provides evidence for the second aerosol indirect effect since higher CDNC
suppress rainfall, leading to higher LWC.

Three of the cases observed during the NETCARE 2014 flight campaign had cloud droplet number
concentrations at or below the tenuous cloud regime (Leaitch et al., 2016; Mauritsen et al., 2011).

Termed the Mauritsen limit by Leaitch et al. (2016), it is a proposed threshold for aerosol concentration,
below which cloud droplets that form grow to sizes large enough to precipitate.

In this study, we will attempt to reproduce the observed linear relationship between LWC and CDNC
using the Single Column Model for Arctic Boundary Layer Clouds (SCM-ABLC), which is based on

the fourth generation of the Canadian Atmospheric Global Climate Model (CanAM4) (von Salzen et al.,
2013). Specifically, we will examine whether autoconversion can explain the observed linear
relationship between CDNC and LWC, since the SCM-ABLC does not include radiative feedbacks
involved in increasing condensation rates or parameterizations of inhomogeneous mixing, relying on a
first order turbulence closure. Dry air above the cloud is allowed to mix into the cloud and evaporate

cloud droplets, but this parameterization may not be sufficient to accurately account for the effect of
stirring between cloudy and non-cloudy air, resulting in inhomogeneous mixing (Gerber et al., 2008;
Jensen et al., 1985). The simulated CDNC and LWC using three autoconversion schemes (Wood,
2005b; Liu and Daum, 2004; Khairoutdinov and Kogan, 2000) will be explored and compared. We will
also examine if the differences from the autoconversion schemes significantly change the radiative

balance of the simulated cloud by using an offline version of the radiative transfer model in CanAM4.3
(see Section 2.3 for details).

## 2. Methods

### 2.1. Observations

This study uses observations from the Network on Climate and Aerosols: Addressing Key
Uncertainties in Remote Canadian Environments (NETCARE) project (Abbatt et al., 2019). These data
were collected during an aircraft campaign on board the Alfred Wegener Institute's Polar 6 aircraft
based out of Resolute Bay, Nunavut (74º40′48″ N, 94º52′12″ W), in July 2014 (see Figure 1). Only
details relevant to this study are included below. A more extensive description of the details of the

flight campaign can be found in Leaitch et al. (2016).



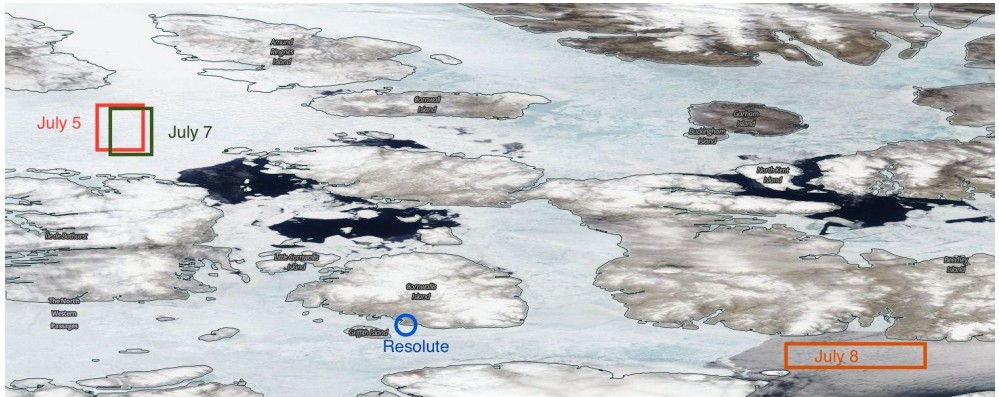

Figure 1. Satellite image from July 8, 2014 depicting Resolute Bay and the surrounding area, with
rectangles showing the approximate locations of profiles on July 5, 7, and 8. Retrieved from
https://worldview.earthdata.nasa.gov/

Temperature, wind speed, and relative humidity measurements from the Aircraft Integrated

Meteorological Measurement System (AIMMS-20) were used in the creation of input profiles for the

SCM-ABLC. Cloud properties were determined from the Forward Scattering Spectrometer Probe

(FSSP-100, Particle Measuring Systems), which measured the number concentration and size

distribution of cloud droplets, allowing the LWC and CDNC to be determined. The FSSP was mounted

in a canister under the port-side wing (Leaitch et al., 2016), with modified tips to reduce shattering

artifacts as per Korolev et al. (2011). These data were processed to account for the geometry of the

FSSP (depth of field = 0.298 cm, beam diameter = 0.02 cm and the true air speed from the AIMMS-20).

No corrections were applied for probe dead-time or for coincidence effects since these were deemed to

be negligible due to the low airspeed of the aircraft (~65 m/s) and low CDNC (< 131/cm$^3$) in this study,

respectively. However, LWC may be underestimated due to droplets that were larger than the upper

limits of the chosen FSSP sampling sizes, which were sometimes set below the upper detection limit of

the FSSP. It is also possible that some droplets were larger than the actual upper detection limit of the

FSSP of 45 µm (Leaitch et al., 2016). However, we expect that the number of larger droplets was

negligible in this work.

### 2.1. Vertical Profiles

Flight sections through and near low clouds (defined as cloud top height ≤ 220 metres) from July 5, 7,

and 8, 2014 were included in this study and the profile locations and times chosen are shown in Table 1.

Each profile contains a single trip either up or down by the aircraft and were chosen for segments when



observations existed for at least 20 m in and above the cloud. Additionally, data points were excluded
when any one of the instruments collecting the data that went into the input profiles malfunctioned. As
many profiles as possible from the Leaitch et al. (2016) study were included in this study. However,
profiles either through very thin cloud layers or entirely within a cloud layer without any observations
above the cloud were excluded.

| Date July 2014 | Start Time (UT) | End Time (UT) | Lowest Cloud Altitude Bin (m) | Highest Cloud Altitude Bin (m) | Mean CDNC ($/cm^3$) | Starting Latitude | Ending Latitude | Starting Longitude | Ending Longitude |
|---|---|---|---|---|---|---|---|---|---|
| 5 | 16:17:09 | 16:18:31 | 100 | 130 | 5.5 | 77.3284 | 77.2796 | -98.7378 | -98.8190 |
| 7 | 16:20:54 | 16:26:58 | 90 | 150 | 15 | 77.1818 | 77.3280 | -98.4485 | -98.8793 |
| 7 | 16:26:59 | 16:28:54 | 80 | 110 | 17 | 77.3273 | 77.2580 | -98.8786 | -98.7206 |
| 8 | 17:27:20 | 17:29:02 | 140 | 190 | 96 | 74.1878 | 74.1895 | -87.8455 | -88.0827 |
| 8 | 17:29:03 | 17:29:57 | 150 | 200 | 87 | 74.1895 | 74.1916 | -88.0851 | -88.2086 |
| 8 | 17:31:29 | 17:32:16 | 150 | 190 | 70 | 74.2006 | 74.2046 | -88.4050 | -88.5083 |
| 8 | 17:32:17 | 17:33:00 | 150 | 200 | 49 | 74.2047 | 74.2090 | -88.5105 | -88.6061 |
| 8 | 17:35:00 | 17:35:43 | 150 | 190 | 100 | 74.2313 | 74.2401 | -88.8686 | -88.9604 |
| 8 | 17:35:44 | 17:36:22 | 150 | 210 | 114 | 74.2403 | 74.2471 | -88.9626 | -89.0419 |
| 8 | 17:38:25 | 17:39:12 | 150 | 220 | 105 | 74.2712 | 74.2816 | -89.3039 | -89.4023 |
| 8 | 17:43:29 | 17:44:43 | 150 | 200 | 93 | 74.3361 | 74.3520 | -89.9603 | -90.1210 |

Table 1. Details of the location and time of the low clouds examined in this study.


Our model represents spatially-averaged conditions in cloudy and clear-sky grid cells separately for a
better comparison with observations, so non-cloudy samples were removed before averaging data
points in cloudy grid cells. This was accomplished by binning LWC data points in each profile into 10
metre altitude bins. Bins were then categorized as being in cloud if more than 50% of the LWC data
points were greater than 0.01 $g/m^3$. For bins deemed to be in cloud, only individual data points within
each in-cloud bin with LWC greater than 0.01 $g/m^3$ were included in the bin's average LWC. A similar
procedure was applied to altitude bins considered to be out of cloud, but with a condition that the





average and individual LWC had to be less than 0.01 g/m$^3$ in order to be included. Meteorological
variables were also averaged into altitude bins, but only observations associated with LWC values
included in the bin average were included in the analysis.

The SCM-ABLC only used a single input of CDNC for each profile. As such, a mean CDNC was
calculated throughout the observed portion of the cloud by averaging the CDNC corresponding to each
LWC data point in the in-cloud altitude bins over the number of data points in that bin. An average
over all of the in-cloud altitude bins was then calculated and used as a fixed input in the SCM-ABLC.
This two-step averaging procedure accounted for potential bias from the length of time the aircraft flew
at each altitude.

## 2.2. SCM-ABLC

*2.2.1 Cloud Physics and Processes*

Much of the model physics of the SCM-ABLC, from cloud processes and turbulence to the
parameterizations of the ocean surface, is taken from the Canadian Atmospheric Global Climate Model,
CanAM4 (von Salzen et al., 2013). However, the SCM-ABLC only models liquid clouds, excluding ice
and mixed-phase clouds, and does not include aerosol processes. Clouds are produced by local
turbulent mixing processes, which move moisture, heat, and momentum down-gradient, and are
affected by surface fluxes. Cloud microphysical processes are prognostic using a scheme based on the
governing equations for water vapour and cloud liquid water outlined in Lohmann and Roeckner (1996)
and Lohmann (1996) (von Salzen et al., 2013).

Eddy diffusivities calculated in the model depend on horizontal wind, height above ground, the
gradient Richardson number, and a mixing length (von Salzen et al., 2013). In the presence of cloud,
the mixing length is set to 100 metres (von Salzen et al., 2013), while in the absence of cloud, the
mixing length is calculated from the parameterization by Lenderink and Holtslag (2004). Surface fluxes,
including evaporation from the ocean, as well as heat and momentum fluxes, are simulated using an
approach based on Monin-Obukhov similarity theory (von Salzen et al., 2013).

The vertical size of the grid cells in SCM ABLC is 10 m, which allows for a straightforward
comparison with the flight observations since they are over a relatively narrow period in time and space
with high temporal resolution (see Table 1) so that vertical features of the clouds are resolved on scales



of a few metres. The modelled lower boundary was the ground, but the height of the upper boundary varied with the cloud top height and availability of measurements (see Table 1 for cloud top heights), though the upper boundary was always at least 20 m above the observed cloud top. The time step used was 900 seconds. The total run time was 300 hours, which ensured that model results approach equilibrium for the given boundary conditions.


Unsaturated air can be entrained into the cloud at the top and sides of the cloud as well as the bottom and affect microphysical properties in the cloud (e.g. Gerber et al., 2008). Entrained parcels have been found to exist on scales of meters in size, and can reach up to tens of meters into the clouds before mixing homogenizes them with the rest of the cloud (Gerber et al., 2008). In the model, cloud

parameterizations do not account for lateral mixing. While our July 8 flight observations are unlikely to have many entrained parcels due to the horizontal extent of the cloud, observations on July 5 and 7 are more likely to contain entrained parcels. Similar to other large-scale atmospheric models, air mixed into the cloud by vertical diffusion at the top and bottom of the cloud is immediately mixed with the cloudy air assuming horizontally uniform thermodynamic cloud properties and cloud droplet number

concentration.

### 2.2.2 Input profiles and boundary conditions

Inputs to the SCM-ABLC used aircraft observations of wind speed, relative humidity, LWC, CDNC, and temperature. These inputs provided initial conditions for the model. Additionally, mean vertical

profiles of CDNC, temperature, specific humidity, and horizontal winds for each individual aircraft ascent or descent are generated and used to constrain meteorological conditions in the simulation by nudging (see Supplement). Upper boundary conditions for cloud simulations representing the bottom of the free troposphere based on aircraft measurements were nudged as to remain constant over the duration of the model run. The lower boundary conditions at 10 m height for temperature and pressure

were specified: the surface temperature was set to 273 K as the flights were all near or over open water and ice edges and the surface pressure was set to 1013 hPa. Between the surface and the altitude of the lowest observation-based initial condition, LWC, horizontal wind, and temperature were calculated based on vertical diffusion with a first order turbulence closure (von Salzen et al., 2013). Model output from the layers beneath the cloud were not considered in the analysis of results in the following

sections.





### 2.2.3 Boundary Layer Heights

The choice of model domain vertical extent is important in the SCM-ABLC since processes above the boundary layer are not well represented in the model due to the relatively long time scales and non-

local character of free and upper tropospheric processes. For instance, the model does not account for the large-scale transport of air. On the other hand, mixing processes and cloud microphysical processes occur on time scales that are fast compared to large-scale transport of air so that it is sufficient to relax large-scale simulated thermodynamic conditions towards observed profiles. Consequently, we assume that the free troposphere in the model can be represented by the observations at those heights, and

properties remain constant over the time period of the profile. The boundary layer height was estimated from the height of the base of the observed temperature inversion. The modelled boundary layer height was then varied within 30 m of the height of the base of the inversion such that the simulated LWC profile produces a maximum at an altitude similar to the observed profile.

*2.2.5 Autoconversion*

Three autoconversion schemes detailed in the literature were used in the SCM-ABLC – Wood (2005b), Liu and Daum (2004), and Khairoutdinov and Kogan (2000). The latter two are herein abbreviated as L&D and K&K, respectively. These schemes are described below.

The autoconversion scheme presented by Khairoutdinov and Kogan (2000) separates liquid water in the model into two categories: cloud liquid water and drizzle. It predicts drizzle water and drizzle drop concentration using a prognostic scheme by fitting results from a large-eddy scheme model (Khairoutdinov and Kogan, 2000). This scheme was found to be in good agreement with an explicit model for two cases with no rain and heavy drizzle that were analyzed by Khairoutdinov and Kogan

(2000). It was developed for conditions found in the extra-tropics and midlatitudes off the west coasts of continents where stratocumulus cloud layers arise from upwelling of cold water in the ocean(Khairoutdinov and Kogan, 2000). As in the CanAM4, the K&K scheme in the SCM-ABLC has been tuned so that the rate of conversion from cloud droplets to rain drops has been increased relative to the original parameterization (von Salzen et al., 2013). Tuning factors are commonly used in climate

models as autoconversion is usually underestimated due to missing processes and other factors (e.g. cloud homogeneity) (Williamson et al., 2015). A tuning factor of 2.5, based on simulations with version 4.3 of the Canadian Atmospheric Model (CanAM4.3), is used in this paper.



The scheme by Liu and Daum (2004) is based on the similar principles as K&K, but does not assume a fixed collection efficiency with respect to droplet radius (Liu and Daum, 2004). The better representation of the physics involved in the L&D autoconversion scheme results in stronger dependencies on LWC and cloud droplet number concentration (Liu and Daum, 2004). It also increases the coefficient of variation (the ratio of standard deviation to the mean radius), which affects the threshold radius for autoconversion as broader droplet size distributions tend to have larger

autoconversion rates (Liu and Daum, 2004). Unlike K&K, L&D has a threshold radius value before autoconversion begins, preventing rain processes below the threshold. However, this scheme has been shown to overestimate the autoconversion rate above the threshold compared to observation-based estimates for mid-latitude marine clouds (Wood, 2005b).

The Wood (2005b) scheme reduced the constant term in the L&D parameterization to 12% of its original value based on a comparison with observation-based autoconversion rates in drizzling stratiform clouds that showed lower rates than predicted by L&D. Wood (2005b) also found that the K&K scheme did not over-predict rain as much as the L&D scheme in test cases (flight data described in Wood 2005a), and suggested that the K&K scheme may be useful in situations other than those it

was designed for (Wood 2005b). The modified L&D scheme (referred to as the Wood scheme) produced more realistic dependencies of autoconversion on cloud LWC and CDNC compared to the original L&D scheme for drizzle in stratiform clouds (Wood 2005b). All three of these schemes have been used in various modelling applications and were originally developed for the mid-latitudes. As part of our study, we will be evaluating their performance to summer Arctic low clouds.


Three additional cases were simulated in the SCM-ABLC for diagnostic purposes. The first two cases eliminated the impacts of CDNC on the autoconversion rates. This was accomplished by keeping the CDNC constant while retaining the variation in meteorological conditions, such as temperature, relative humidity, and wind speeds. CDNC values of $5/cm^3$ and $112/cm^3$, near the extreme observed values,

were chosen to represent the range from the observations caused by CDNC. Only the Wood autoconversion scheme was used for these calculations for simplicity. The third case simulated no autoconversion by allowing the variable that represents rain water to be constantly zero, forcing all of the moisture in the clouds to remain in either cloud droplet or vapour form.

**2.3. Offline Radiative Transfer Model**





In addition to SCM-ABLC, this study uses an offline version of the radiative transfer model in CanAM4.3. The main attributes of the radiative transfer model are described in von Salzen et al. (2013) and references therein. Only profiles from July 8 are used for the radiative transfer calculations. The flights from July 5 and 7 were not analyzed due to the different solar zenith angles and the possible

effects of a different regime at lower CDNC. We summarize the most important aspects of the model below.

### 2.3.1 Model Description

Solar and infrared fluxes and heating rates are computed using the Monte Carlo Independent Column

Approximation (McICA), which can account for the cloud horizontal variability and vertical overlap (Pincus, Barker, and Morcrette, 2003; Barker et al., 2008). Both the solar and infrared use two-stream solutions, the delta-Eddington approximation for the solar (Zdunkowski et al., 1982), and a perturbation approach for the infrared (Li, 2002).

Absorption by gases is computed using the correlated-$k$ method (von Salzen et al., 2013; Lacis and Oinas, 1991). The optical properties of liquid cloud and aerosols are computed using the parameterizations referenced by von Salzen et al. (2013).

Aerosols were omitted in the radiative transfer calculations due to their relatively small effects on the

radiative fluxes compared to those due to the clouds.

### 2.3.2 Cloud Profiles

The radiative transfer model required profiles of cloud properties including the effective radius, liquid water path (LWP), cloud fraction, and cloud heights. These profiles were constructed by using model

output of cloud properties starting at the top of the simulated cloud down to an altitude that resulted in a cloud thickness equal to the penetration depth of the aircraft into the cloud during the NETCARE flights. Clouds below the lowest flight level of the aircraft were omitted to avoid only relying on model output in all but one of the simulations with the radiative transfer model. We ran a single case using averaged cloud microphysical properties from the observed part of the cloud in order to estimate the

difference in radiative fluxes due to the difference in cloud thickness (see Section 3.3 for results). The LWC was then multiplied by the grid cell depth and integrated to yield the LWP needed as input to the radiative transfer model. The cloud amount was set to 1 (overcast) at the altitudes where there was



cloud, for both the SCM-ABLC and observed profiles. This allowed the optical depths of the modelled
and observed clouds to be compared since their thicknesses were equal.


The radiative transfer calculations were performed using the cloud profiles constructed as described
above using three configurations: cloud profiles from observations, cloud profiles from the SCM-
ABLC, and no clouds. The profile with no clouds was calculated by inputing zero values for the cloud
amount, LWP, and effective radii. The radiative effects of clouds were computed by subtracting the
clear-sky radiative fluxes from the radiative fluxes resulting from cloudy profiles.

*2.3.3 Atmospheric State Profiles*

Profiles of pressure, temperature, and water vapour profiles were created using the European Centre for
Medium-Range Weather Forecasts (ECMWF) Re-Analysis (ERA)-Interim product by extracting the
profiles closest in time and location to the aircraft profiles. The results were vertically interpolated to a
vertical grid with 8866 levels from the surface to ~89 km with each layer between 10 and 20 metres
thick. The temperature profiles from ERA-Interim were adjusted by a height-independent scaling factor
defined by comparing the mean cloud temperatures from the ERA-Interim to the mean observed cloud
temperatures, bringing the cloud temperatures closer to the observations.  The surface skin temperature
was chosen by rounding the temperature interpolation at the lowest level to the nearest degree. Trace
gas profiles, including carbon monoxide, carbon dioxide, ozone, nitrous oxide, methane, oxygen,
carbon tetrachloride, CFC-11 and CFC-12, were computed by interpolating a climatology from the
ECMWF Integrated Forecasting System to all levels.

*2.3.4 Surface Albedo*

The flight on July 8 took place over the open ocean, which we estimated to have a broadband surface
albedo of 0.054 based on the solar zenith angle and the time of flight using the parameterization from
Taylor et al. (1996). This value is consistent with ocean albedos used by other studies based on
measurements (Henderson-Sellers and Huges, 1982; Kukla and Robinson, 1980; Budyko, 1956; Payne,
1972). Albedo values from July 5 and 7 were not used, as the profiles from July 5 and 7 are omitted
from the radiative transfer calculations.

## 3. Results and discussion



The green triangles in Figure 2 show the observed mean LWC and CDNC from the profiles listed in

Table 1 and Section 2.1.2. As expected from Leaitch et al. (2016), the CDNC and LWC are linearly

related, despite a slightly different definition of profiles. The variance in our observed relationship is

low, with $R^2 = 0.987$ (Table 2, "Observed" case).

To determine whether autoconversion was an important driver of the linear relationship observed

between LWC and CDNC by Leaitch et al. (2016), we used the SCM-ABLC to model the LWC for the

profiles listed in Table 1 using the three different parameterizations of autoconversion (described in

Section 2.2.5). Simulations were conducted with the K&K, L&D, and Wood autoconversion schemes,

with two different constant CDNC, and with no autoconversion scheme (see Figure 2 and Table 2 for

the cases K&K, L&D, Wood, All CDNC 112/cm$^3$, All CDNC 5/cm$^3$, and No Rain, respectively). Based

on these results, we constructed a linearization of the closest-fitting results to observations, called

"L&D and K&K" as it corresponds to the combination of L&D and K&K schemes so as to use the

K&K scheme at CDNC < 20/cm$^3$ and the L&D scheme at higher CDNC.

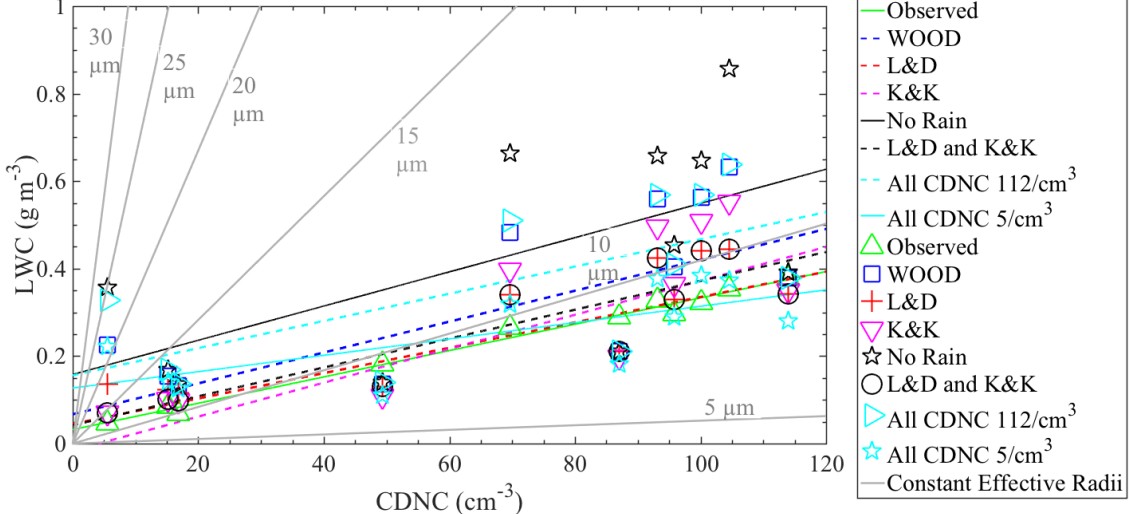

Figure 2. Observed and simulated LWC for three autoconversion schemes in SCM-ABLC, as a

function of the observed CDNC specified in the model (symbols). Linear regressions are also shown

for the observations and different parameterizations (lines). 'No rain' corresponds to the LWC

produced by the model with no autoconversion scheme. 'L&D and K&K' corresponds to the

combination of L&D (>20/cm$^3$) and K&K (<20/cm$^3$) schemes. 'All CDNC 5/cm$^3$' and 'All CDNC



112/cm$^3$' refer to the test cases of the Wood scheme that were run with all of the profiles having
constant CDNC of 5/cm$^3$ and 112/cm$^3$, respectively, with the x-axis values corresponding to which
original CDNC values they had. The grey lines show the LWC for varying CDNC given the constant
effective radii of the labels.

|  | Slope | $R^2$ | Intercept | Mean LWC (g/m$^3$) | Variance |
|---|---|---|---|---|---|
| Observed | 0.00301 | 0.987 | 0.032 | 0.24 | 0.013 |
| Wood | 0.00353 | 0.554 | 0.067 | 0.35 | 0.032 |
| L&D | 0.00290 | 0.736 | 0.045 | 0.28 | 0.017 |
| K&K | 0.00388 | 0.707 | -0.016 | 0.29 | 0.031 |
| No Rain | 0.00391 | 0.387 | 0.158 | 0.43 | 0.057 |
| L&D and K&K | 0.00330 | 0.795 | 0.042 | 0.27 | 0.020 |
| All CDNC 5/cm$^3$ | 0.00187 | 0.512 | 0.127 | 0.25 | 0.010 |
| All CDNC 112/cm$^3$ | 0.00311 | 0.443 | 0.156 | 0.37 | 0.032 |

Table 2. Summary of linear fits of observations and model output. See main text for description of
cases. Here $R^2$ corresponds to the coefficient of determination, or the proportion of variance in LWC
due to CDNC.

Overall, Figure 2 shows that the linearity of the relationship observed between CDNC and LWC can be
reproduced by all three autoconversion schemes. Nevertheless, the tested autoconversion schemes tend
to overpredict the LWC compared to observations in most cases. The Wood scheme (blue squares)
produces the highest variability in LWC and overpredicts the observations the most. The K&K scheme
(magenta triangles) has the largest slope but overpredicts the least at lower CDNC, while the L&D
scheme (red crosses) has the lowest slope and overpredicts the observations the least at higher CDNC.

The slopes and variance in Table 2 show that the L&D scheme is closer to the observations than the
Wood scheme in both measures, suggesting that the reduction in autoconversion implemented by Wood
to the original L&D autoconversion scheme is not suitable for summer Arctic low clouds. In summary,
the simulations with L&D and K&K parameterizations explain most of observed variability in LWC in
Fig. 2.




Interestingly, simulations with the simplified parameterizations that do not account for effects of CDNC on autoconversion ('All CDNC 5/cm$^3$' and 'All CDNC 112/cm$^3$') also produce LWC values that are similar to the observed values for each flight, with just slightly lower R$^2$ values compared to results with L&D and K&K parameterizations (see Table 2). This indicates that differences in

meteorological conditions, cloud top height, boundary layer depth, and the location of the inversion in the simulations that are associated with different aircraft profiles are partly responsible for the increase in LWC with CDNC according to the linear regression in Figure 2. This conclusion is further supported by the results of the simulation which does not include autoconversion and precipitation (the 'No Rain' case). Without autoconversion and precipitation, the simulated LWC is generally much higher than

observed, but high values of LWC are still associated with high observed CDNC (see Figure 2). The 'No Rain' case has a larger slope and smaller R$^2$ than the other test cases, supporting the hypothesis that autoconversion is an important contributor to the observed linearity between LWC and CDNC compared to the other processes represented by the model. However, the relatively small number of flight profiles substantially affects the robustness of the statistical relationship between CDNC and

LWC. Consequently, the model results indicate that a larger number of measurements would be required in order to minimize the impact of meteorological variability on LWC and relationship with CDNC.

Overall, the K&K scheme reproduced the observed LWC better at CDNC below 20/cm$^3$ while the

L&D scheme reproduced it better at higher CDNC, suggestive of a regime change like that described by Mauritsen et al. (2011). Below the Mauritsen limit, clouds are CCN-limited and any droplet that forms can drizzle out. This process seems to be better represented by the K&K scheme which continuously converts cloud droplets to rain drops with no threshold for conversion, compared to the other schemes which have a constant threshold, i.e the L&D and Wood schemes. At higher CDNC, the

K&K scheme overpredicts the LWC compared to the L&D scheme. To capture this change in regime, we combined the L&D and K&K schemes by using the K&K scheme to model the three profiles with CDNC below 20/cm$^3$ and the L&D scheme for the rest. This combination performed the best at obtaining the lowest variance and the overall slope is similar to the observations (Table 2, "L&D and K&K" case). The exact cut off for the tenuous cloud regime is debatable. In the original observations

of Mauritsen et al. (2011), they discussed a threshold of 10/cm$^3$. However, both Mauritsen et al. (2011) and Leaitch et al. (2016) suggested that this limit is more reflective of a change in regime than a specific numerical cutoff and that the actual threshold depends on location and time. The three lowest




mean CDNC values used in our modelling were all less than or equal to 17/cm$^3$, similar to the limit suggested by Leaitch et al. (2016) of 16/cm$^3$. We stress, however, that our data set is limited to only

three profiles with CDNC in the tenuous cloud regime and further work would be needed before these results could be generalized. Nevertheless, our findings are consistent with the observational results from Mauritsen et al. (2011) and Leaitch et al. (2016), and further demonstrate the possible importance of this regime change at low CDNC. Other models may also need to consider this regime change to better represent Arctic low clouds.


The two observed profiles for which the model consistently underpredicted the LWC (at CDNC concentrations of 49/cm$^3$ and 87/cm$^3$) had lower wind speeds in the cloud and less of a difference in wind speeds between in-cloud and above the cloud than some of the other profiles. This may have prevented sufficient water vapour from mixing into the cloud, thereby preventing conversion of cloud

water vapour to liquid water.

Other studies have previously noted that autoconversion schemes often do not represent the rain rates in the Arctic very well (Croft et al., 2016; Zhang et al., 2002; Olsson et al., 1998). Olsson et al. (1998) speculated that the discrepancy between modelled and observed rain rates may be due to the size of

droplets, as small droplets can fail to initialize autoconversion when the threshold is too large. Our results support this theory at low CDNC: the K&K scheme, which has no threshold for autoconversion, performs the best at low CDNC, suggesting that the thresholds for autoconversion may be too high in the L&D and Wood schemes at these droplet concentrations, resulting in overpredicted LWC. We found that the L&D scheme does best at higher CDNC, so there may be a regime change between low

and high CDNC. Although the model comparisons carried out by Stevens et al. (2018) did not directly compare autoconversion schemes, they demonstrated that both large-eddy simulation and numerical weather prediction models showed pronounced tendencies to increase LWP with increasing CDNC, and that LWP is highly sensitive to CDNC, consistent with our results.

Although the L&D scheme best reproduces the nearly linear relationship between the observed LWC and CDNC, the linearity appears to be well-reproduced by all three of the autoconversion schemes that we examined. This indicates that autoconversion is indeed an important driver of the linearity between LWC and CDNC, as the case with no autoconversion is much less linear and with lower R$^2$ (see Figure 2 and Table 2). Since the linear fit for the 'No Rain' case explains less of the variability than the linear



fits for the simulations with autoconversion parameterizations, we surmise that autoconversion is a

driver of the linear relationship. As such, autoconversion appears to be sufficient to drive the linearity

observed between LWC and CDNC by Leaitch et al. (2016), based on our modelling. This is consistent

with the second aerosol indirect effect, and similar to the findings by Stevens et al. (2018). However,

relationships between CDNC and LWC were previously analyzed and explained in other studies,

including Gerber et al. (2008), Albrecht (1989), and Jensen et al. (1985). From these studies, it is clear

that there may be strong correlations between CDNC and LWC due to effects of turbulent mixing and

evaporation of cloud droplets, depending on the efficiency of mixing versus evaporation. However,

there is no evidence of strong turbulent mixing in the observations. Further, we are assuming that

turbulence affects the LWC but not the CDNC in the simulations. We also do not account for cloud

inhomogeneities. As such, the simulated relationship between LWC and CDNC may be incomplete.

Future generations of modellers may have to think about how to better incorporate subgrid-scale cloud

mixing processes in models.

## 3.3. Radiation

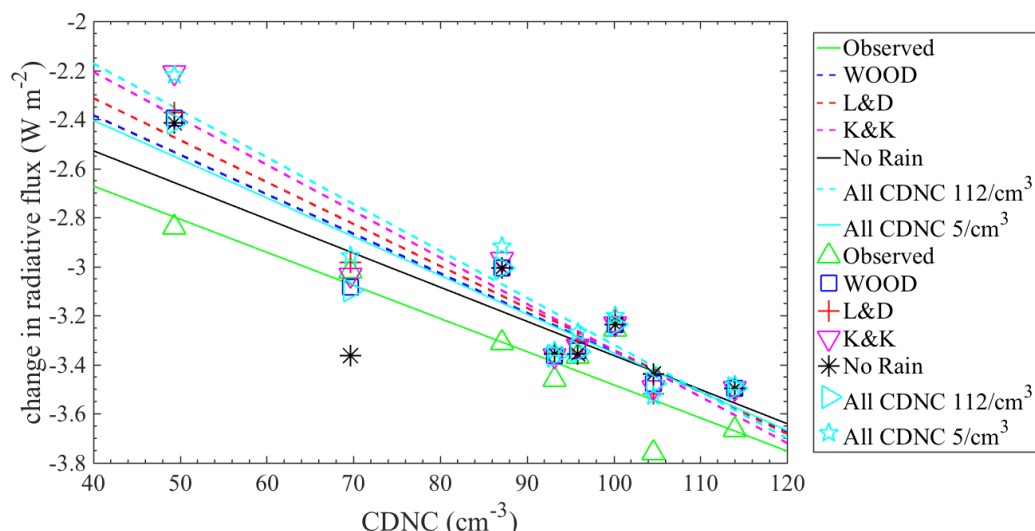


Figure 3. Change in upward longwave radiation at the top of the atmosphere due to the presence of
cloud, on July 8 only, wherein the input cloud variables were from the SCM-ABLC output or based on
observations. The radiative flux for the 'observed' case is calculated using the radiative transfer model
with cloud inputs from observations.




The offline radiative transfer model was run using simulated profiles of liquid water path and effective radius from the SCM-ABLC as input for the flights on July 8, as well as with the clouds removed, to compute the cloud radiative effect. For these calculations, all profiles were assumed to be over open ocean. The green triangles in Figure 3 are the longwave radiative fluxes at the top of the atmosphere
calculated from the observed liquid water path and effective radius, while the other symbols represent the longwave radiative flux calculated from the model output using the different autoconversion schemes in the SCM-ABLC. Since the effective radii are roughly constant over all of the cases that were considered and the LWC was found to linearly increase with the CDNC, the optical depth, and therefore the extinction, estimated from the plane-parallel approximation, also varies linearly at these
relatively low CDNC (see Table 3). This results in the longwave radiative flux at the top of the atmosphere linearly decreasing with increasing CDNC. We find that slopes are slightly larger for the simulations than for observations, with the exception of the "No Rain" case (see Table 3). The $R^2$ value indicates that the relationships are linear to a very good approximation for each case, but lowest for the "No Rain" case (see Table 3). Further, using a t-test, the longwave calculations showed no significant
difference at $p=0.05$ in the radiative effect due to the cloud between the different autoconversion schemes (see Table 4). However, there is a significant difference between the radiative calculations due to the clouds modelled and those based on observations at $p=0.05$ due to the differences in modelled and observed effective radii and LWC for all autoconversion schemes except for the "No Rain" case where no autoconversion was included (see Table 4).


|  |  | Slope | Intercept | $R^2$ |
|---|---|---|---|---|
| Upward longwave flux at the top of the atmosphere | Observed | -0.0135 | -2.1317 | 0.832 |
|  | Wood | -0.0161 | -1.7391 | 0.857 |
|  | L&D | -0.0171 | -1.6294 | 0.898 |
|  | K&K | -0.0189 | -1.4509 | 0.861 |
|  | No Rain | -0.0139 | -1.9720 | 0.658 |
|  | All CDNC 5/cm$^3$ | -0.0158 | -1.7719 | 0.853 |
|  | All CDNC 112/cm$^3$ | -0.0191 | -1.4080 | 0.877 |
| Downward shortwave flux at the surface | Observed | -1.6829 | -20.3366 | 0.770 |
|  | Wood | -2.0970 | 10.0677 | 0.668 |
|  | L&D | -2.0583 | 15.4632 | 0.590 |
|  | K&K | -2.1001 | 25.5905 | 0.721 |



| | | | | |
|---|---|---|---|---|
| | No Rain | -2.0547 | 1.06990 | 0.537 |
| | All CDNC 5/cm$^3$ | -1.9961 | 6.2135 | 0.654 |
| | All CDNC 112/cm$^3$ | -1.7445 | 9.2692 | 0.756 |
| Downward longwave flux at the surface | Observed | 0.30527 | 41.90649 | 0.865 |
| | Wood | 0.43052 | 31.38566 | 0.696 |
| | L&D | 0.43937 | 28.17262 | 0.818 |
| | K&K | 0.50730 | 22.55339 | 0.751 |
| | No Rain | 0.40879 | 34.63778 | 0.590 |
| | All CDNC 5/cm$^3$ | 0.46542 | 23.38898 | 0.772 |
| | All CDNC 112/cm$^3$ | 0.41759 | 32.78588 | 0.674 |

Table 3. Summary of linear fits of radiation model output. See main text for description of cases. Here $R^2$ corresponds to the coefficient of determination, or the proportion of variance in the change in radiation due to the presence of cloud due to CDNC.


| | | L&D | K&K | 5/cm$^3$ | 112/cm$^3$ | No Rain | Observed |
|---|---|---|---|---|---|---|---|
| Upward longwave flux at the top of the atmosphere | WOOD | 0.237 | 0.169 | 0.047 | 0.147 | 0.378 | 0.036 |
| | L&D | - | 0.500 | 0.060 | 0.196 | 0.323 | 0.013 |
| | K&K | - | - | 0.091 | 0.165 | 0.180 | 0.038 |
| | All CDNC 5/cm$^3$ | - | - | - | 0.050 | 0.120 | 0.016 |
| | All CDNC 112/cm$^3$ | - | - | - | - | 0.423 | 0.038 |
| | No Rain | - | - | - | - | - | 0.184 |
| Downward shortwave flux at the surface | WOOD | 0.006 | 7.79E-5 | 4.11E-4 | 0.027 | 0.023 | 0.976 |
| | L&D | - | 0.137 | 8.90E-5 | 0.005 | 0.011 | 0.045 |
| | K&K | - | - | 0.003 | 1.89E-4 | 0.004 | 0.352 |
| | All CDNC 5/cm$^3$ | - | - | - | 3.50E-4 | 0.002 | 0.007 |
| | All CDNC 112/cm$^3$ | - | - | - | - | 0.027 | 0.886 |
| | No Rain | - | - | - | - | - | 0.386 |
| Downward longwave flux at the surface | WOOD | 0.010 | 0.012 | 2.24E-5 | 0.061 | 0.034 | 0.800 |
| | L&D | - | 0.629 | 4.32E-4 | 0.009 | 0.015 | 0.389 |
| | K&K | - | - | 8.66E-4 | 0.014 | 0.007 | 0.648 |
| | All CDNC 5/cm$^3$ | - | - | - | 5.11E-5 | 3.55E-4 | 0.108 |
| | All CDNC 112/cm$^3$ | - | - | - | - | 0.041 | 0.729 |
| | No Rain | - | - | - | - | - | 0.513 |





Table 4. t-test results for the change in radiative flux due to the presence of cloud for July 8.

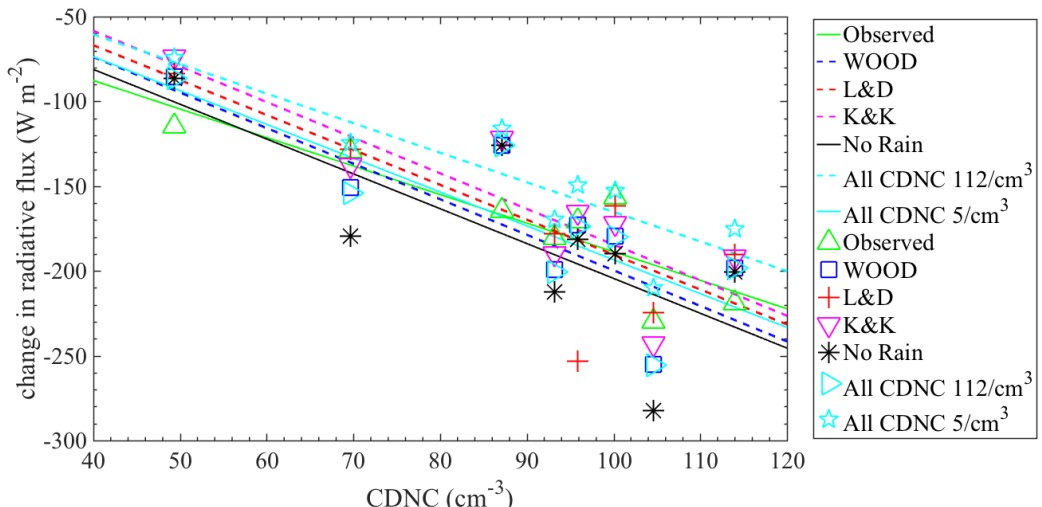

Figure 4. Change in downward shortwave radiation at the surface due to the presence of cloud, wherein the input cloud variables were from the SCM-ABLC output or based on observations. The radiative flux for the 'observed' case is calculated using the radiative transfer model with cloud inputs from observations.

A similar decreasing linear relationship exists for the downward shortwave radiation at the surface (Figure 4). However, there is no significant difference at $p=0.01$ (see Table 4) in the downward shortwave radiative effect at the surface between each scheme and the observation-based model runs on July 8 except for the "All CDNC 5/cm$^3$" case.





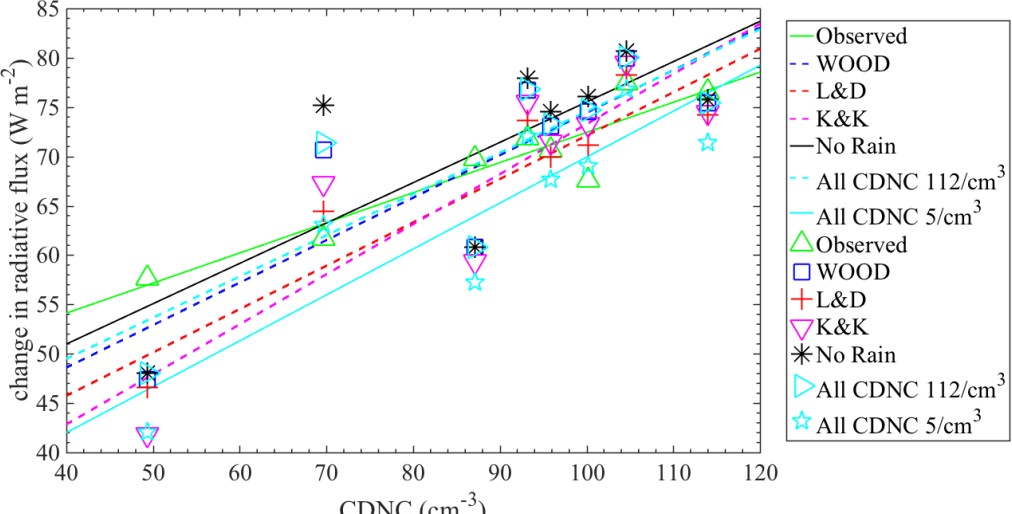


Figure 5. Change in downward longwave radiation at the surface due to the presence of cloud, wherein the input cloud variables were from the SCM-ABLC output or based on observations. The radiative flux for the 'observed' case is calculated using the radiative transfer model with cloud inputs from observations.


An increasing linear relationship exists for the downward longwave radiation at the surface (Figure 5), indicating that clouds with higher CDNC result in greater longwave radiative fluxes when compared to the case with no cloud. The calculation based on observations results in the highest $R^2$ value (see Table 3), implying that autoconversion schemes do not replicate this result quite as well, although the L&D

scheme does quite well at linearizing despite having a very different slope and intercept (Table 3). T-tests show, however, that none of the autoconversion schemes result in downward longwave radiation values that differ significantly (p=0.01) from observation-based calculations, though the "All CDNC 5/cm$^3$" case and the "No Rain" case differ significantly (p=0.05) from all other autoconversion-based cases (Table 4).


From these comparisons of the July 8 data, the most important result is that there is an offset in the radiative calculations based on the observations versus the SCM-ABLC model output for the upward longwave radiative flux at the top of the atmosphere which is significant at p=0.05 for all cases but "No Rain," which had no autoconversion processes. However, the downward shortwave radiative flux at the

model surface shows that all autoconversion schemes but "All CDNC 5/cm$^3$" produce fluxes that are





not significantly different (p=0.01) from those calculated based on observed cloud profiles. A final
takeaway from the t-tests was that the Wood autoconversion produced small but statistically
significantly different downward shortwave radiation at the surface from the other two autoconversion
schemes from the literature at a significance of p=0.01, while the L&D and K&K schemes did not

significantly differ from each other. This may require further investigation as the L&D and Wood
schemes differ only by a constant, while the K&K scheme differs significantly from those two. In
addition to this, the "No Rain" case with no autoconversion processes differed significantly at p=0.05
from all other SCM-ABLC-based input to the downward shortwave flux at the surface, so the presence
of an autoconversion scheme in the cloud model produces a significant change in the results of

radiative modelling.

A sample calculation was carried out to test the radiative effects of extending the cloud to the surface,
as was surmised to occur by observers during the July 8 flight. The extension of the cloud was assumed
to have a LWP and effective radii equal to the average of those values in the observed portion of the

cloud. This resulted in a decrease of less than 1% in the longwave radiative flux at the top of the
atmosphere. Similarly, the change in the downward longwave flux at the surface was also small, with
the newly modelled cloud increasing the radiative flux by almost 4%. The results were most sensitive
in the downward shortwave flux at the surface, with the thicker cloud decreasing the original flux by
approximately 35%. The small changes in the longwave radiation indicate that the temperatures of the

ground and the cloud top are similar. The larger change in the downward shortwave flux at the surface
indicates that the observed portion of the cloud is insufficiently thick to estimate the effect on the
shortwave flux on its own.

Overall the results from Table 3 show that the model-based radiative flux calculations produce more

negative slopes than those based on observations, suggesting that the model overestimates the
relationship between CDNC and shortwave radiative flux, and that the first aerosol indirect effect may
be overestimated by the parameterizations examined from the literature. The first aerosol indirect effect
depends on a realistic sensitivity of fluxes to changes in CDNC in response to changes in CCN
concentrations. However, the best agreement in slopes for the change in shortwave radiative flux is

found in the simulations which assume a constant CDNC in parameterizations of autoconversion (see
Table 3). In particular, the change in shortwave flux is much greater for the K&K parameterization than
calculation based on observations, which is consistent with the particularly strong non-linear





dependency of this parameterization on CDNC, indicating that aerosol indirect effects may be
especially overestimated with this parameterization.


### 4. Conclusion

Our model simulations show that the linear relationship between LWC and CDNC observed by Leaitch
et al. (2016) in summer Arctic low clouds is consistent with parameterizations of autoconversion,
although other processes, such as variability in meteorological conditions, entrainment of dry air

without mixing and increased condensation rates, may contribute to the observed relationship. The
choice of autoconversion scheme in the SCM-ABLC changes the simulated relationships between
LWC and CDNC, with the best simulated linear relationship (highest $R^2$) obtained from a combination
of the K&K scheme at CDNC below 20/cm$^3$ and the L&D scheme at higher concentrations. These
results are consistent with a regime change between very low and higher CDNC corresponding to the

Mauritsen limit. Below this limit, droplet concentrations are CCN-limited and droplets are expected to
grow and fall out quickly, consistent with the constantly-drizzling K&K scheme. In contrast, the L&D
and Wood schemes have threshold radii before drizzle occurs, consistent with our understanding of
drizzle formation in regions of the world where CDNC are higher. Due to a lack of observational data,
the exact transition above which the L&D scheme performed better could only be constrained to a

range of 17-48/cm$^3$. It is important to note that our observations below the Maurtisen limit only
consisted of 3 profiles and that our conclusions are highly dependent on this limited data set. It would
be of interest to examine whether a regime change can be reproduced with more data and in other
models, as the observational data examined in this study have shown that cloud properties, such as
effective radius, vary somewhat between the regimes, with an average observed effective radius of 12

µm below the Mauritsen limit versus 10 µm above it. The choice of autoconversion scheme is most
relevant when examining the cloud microphysical properties for their own sake, as opposed to radiation,
and the combination of K&K and L&D schemes should be used for these conditions.

The radiative impacts of the modelled downward shortwave and longwave radiation at the surface did

not differ significantly at p=0.01 from those due to the observations using all three autoconversion
schemes from the literature. The radiative impacts of the modelled upward longwave radiation at the
surface did differ significantly from those due to the observations for these schemes at p=0.05. This
suggests that the microphysical parameters such as LWP and effective radius simulated by all the
autoconversion schemes were sufficiently similar to observations for shortwave calculations but not for



upward longwave calculations. The Wood autoconversion scheme simulated downward shortwave

radiation at the surface that was significantly different (p=0.01) from the K&K and L&D schemes,

although not from the observations. This appears to be due to the higher modelled LWC in the Wood

scheme, and indicates that this scheme may be less suitable for modelling low clouds in the summer

Arctic, which tend to have low LWC.


Future work should determine the prevalence of a linear relationship between LWC and CDNC in other

clouds, and whether autoconversion, therefore the second aerosol indirect effect, is one of its primary

drivers. Since part of our results were highly dependent on CDNC below the Mauritsen limit,

determining the prevalence of clouds in a CCN-limited regime is needed to understand the importance

of implementing different autoconversion schemes in clouds. There remain large uncertainties in the

radiative effect of low clouds in the summer Arctic, and ensuring that cloud microphysical properties

are properly represented in models is one way to begin to reduce that uncertainty. Another important

component of reducing the uncertainty in the radiative effect of clouds like these in the summer Arctic

involves comparing the calculated radiative effect to observations. Remote sensing or in-situ

observations would allow us to improve our understanding of modelled cloud radiative effects. These

results could be relevant for other regions with low CDNC such as clean marine clouds and fogs.

*Author contributions*

JD and RYWC wrote this paper. RYWC, KVS, and IF provided project direction and supervision.

RYWC and IF provided funding. KVS and JC provided the majority of the methodology and software.

WRL and GL also contributed methodology. Investigation was primarily carried out by JD, with input

from RM. All authors were involved in the review of this paper.

*Acknowledgements*

We thank the Canadian Centre for Climate Modelling and Analysis (CCCma) for computing time on

their server for running the SCM-ABLC and radiative transfer model. Funding for this work was

provided by the Marine Environmental Observation, Prediction and Response Network (MEOPAR),

which is a federally-funded Networks of Centres of Excellence (NCE) and the Natural Sciences and

Engineering Research Council of Canada (NSERC) through Discovery Grants and the NETCARE

project of the Climate Change and Atmospheric Research Program. NETCARE was also funded by

additional financial and in-kind support from the Alfred Wegener Institute, and Environment and

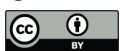



Climate Change Canada, Fisheries and Oceans Canada, and the Major Research Project Management
Fund at the University of Toronto.

We acknowledge the use of imagery from the NASA Worldview application
(https://worldview.earthdata.nasa.gov/) operated by the NASA/Goddard Space Flight Center Earth
Science Data and Information System (ESDIS) project.

*Competing interests*

The authors declare that they have no conflict of interest.

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
