# Peer review of "Modelling the relationship between liquid water content and cloud droplet number concentration observed in low clouds in the summer Arctic and its radiative effects"

_Atmospheric Chemistry and Physics, 2019_

## Referee Comment (RC1) · Anonymous Referee #1 · 7 May 2019

I have reviewed "Modelling the relationship between liquid water content and cloud droplet number concentration observed in low clouds in the summer Arctic and its radiative effects" by Dionne et al. My comments are mainly on presentation, but I do have two substantive concerns on the analysis that put the manuscript in the major revisions category.

[Figure]

**1   Comments on presentation**

I am not an expert on arctic clouds, but globally, the question of LWP adjustments to $N_d$ changes is extremely important in the context of rapid adjustments (formerly the "cloud lifetime effect") to the radiative forcing by aerosol–cloud interactions (formerly the "Twomey effect"). See, e.g., Gryspeerdt et al. (2019 ACP); Rosenfeld et al. (2019 Science); Mulmenstadt and Feingold (2018 Current Climate Change Reports). It might help to make the connection to this literature in the introduction. As currently written, the introduction left me confused whether this is an aerosol–cloud paper (as the focus on $N_d$ would suggest); or a feedbacks paper (as the statement about rapid warming would suggest); or a paper on the interaction between the two, in which case, perhaps cite Nazarenko et al. (2017 JGR) or Lohmann (2017 JGR) or an arctic equivalent, if that exists. However, from the main text, I think it's an ACI paper, so I would focus the introduction on LWP adjustments to the Twomey effect.

The other major presentation question I had after reading the paper was what justified this focus on pure liquid clouds. Perhaps this betrays my ignorance, but I thought the most radiatively important low cloud type was mixed-phase in the Arctic, even in summer (Shupe and Intrieri, 2004?; de Boer et al., 2009).

**2   Comments on the analysis**

My main substantive concern is that I am not convinced the findings are statistically robust. If I understand the analysis correctly, the authors simulated 11 cloud profiles with different mean $N_d$ and LWC. In the observations, cloud-mean LWC is proportional to cloud-mean $N_d$, based on these 11 data points. The authors then tried different microphysical schemes to determine which one is best able to reproduce the observations. A setup without autoconversion gives the worst regression coefficient, and based on

this, the authors conclude that autoconversion is responsible for the proportionality.

My concerns, in detail, are:

1. The no-autoconversion setup does pretty well, actually. It simulates higher LWC than the other setups, but that is to be expected, because a big sink process for LWC is turned off. The slope in Fig. 2 looks indistinguishable from the setups that are claimed to work better.

2. The constant $N_d$ runs also have pretty large slopes, which would indicate to me that a big part of the LWC increase is *not* due to autoconversion, or at least not due to the parameterized $N_d$ dependence in the autoconversion rate.

3. Continuing on that thought: in an attribution of an observed relationship to a candidate process, I would want to see some discussion on why other candidate processes are eliminated. Eliminating candidate processes is, of course, what models excel at, but confirming them, not so much (Oreskes et al. 1994, maybe?). First, I would want to know whether the clouds are adiabatic. Then, I would want to know what stage in their life cycle they are in. At that point, a clearer picture may start to emerge; for example, in an adiabatic cloud, the vertically averaged LWC increases with cloud geometric thickness (thermodynamic conditions being equal), and the geometric thickness increases with $N_d$ (Pincus and Baker, 1994), purely from energy budget considerations.

Based on those concerns, I think a more convincing way to approach the problem would be first to build a conceptual model of the clouds and then to eliminate candidate processes (which probably requires numerical modeling), rather than to pick one process seemingly arbitrarily and trying to "confirm" it (because, as we know, science is the process of hypothesis refutation, not hypothesis confirmation).

My methodological comment is on letting the single-column model run to equilibrium. Actual clouds do not reach equilibrium, because precipitation acts as a condensate

sink that (along with evaporation) causes the clouds to dissipate. In your method, the condensate loss by precipitation is balanced by moisture supply by advection, allowing the cloud to live forever. Clouds that live forever seem like a major limitation in a study on the cloud lifetime effect.

On the other hand, the model obviously needs to spin up.

This is a problem that the authors need to solve, but two suggestions they may find useful are:

1. Argue that clouds at any given point in time are in "quasi"-equilibrium. This is actually an assumption in many GCM parameterizations, i.e., the state the GCM tries to capture is representative of a cloud field averaged over a fairly long time step (30 minutes). However, I don't know if I would buy the argument for an individual cloud.

2. Spin up the model with one $N_d$, then observe the transient behavior when you abruptly change to a different $N_d$.

There is a series of papers by Andrew Gettelman (2015) on SCM studies of different cloud microphysics that might provide insight.

---

## Referee Comment (RC2) · Anonymous Referee #2 · 19 Jun 2019

This paper uses previously published observations (Leaitch ACP2016) of Arctic boundary layers of cloud liquid water content and droplet number to study the response of three autoconversion schemes and then considers the radiative properties of one cloud.

New data analysis demonstrates the linear relationship between cloud drop number concentration and liquid water path for these clouds which is a useful addition to the observational record. A number of samples in cloud were of low droplet number, in the CCN limited regime, referred to as the Mauritsen limit. Only very few samples were

collected in this regime but the linear relationship appears to hold. As mentioned in the text, there is significant variability within the data, perhaps related to background meteorological conditions. It is therefore difficult to ascertain the significance of the result more broadly.

A main focus of the analysis and result is the comparison of three autoconversion schemes with the aim of investigating the impact of autoconversion on the LWP:CDNC relationship. All three of the autoconversion schemes appeared to perform well, suggesting that the process is in fact well constrained, even for the Arctic. The authors then demonstrate that a combination of two schemes, one for the CCN limited regime, and one for larger concentrations of CCN can improve on the performance of a single scheme across the full phase space. Again, the limited data makes this an intriguing but not completely satisfying result, with no attempt to explain, other than to invoke other processes such as turbulence and mixing.

Once the autoconversion schemes have been considered there is a section that investigates the radiative properties of clouds. The paper seems to lead towards a comparison of the radiative properties of clouds that are and are not CCN limited. However, only a cloud that is above this CCN limit is investigate, to the detriment of the work. Further, following the comparison of autoconversion schemes, these are shown to not have a large impact on the radiative properties of the clouds. The finding that the modelled radiation using the autoconversion schemes is different from observations in the July 8th cloud warrants further investigation and may be a useful result. Having only one such case though, is not likely to be sufficient to inform the modelling community of changes that might need to be made to the representation of aerosol indirect effects.

Numerous tables give details of the linear fit parameters, which whilst required, are not so easy to interpret. I would suggest that some measure of the uncertainty / significance is added to the plots to allow the reader to make an informed assessment. This would be useful on Figures 3, 4 and 5. It may also assist the reader to combine those panels in to a single figure. The aims of the paper should be more clearly stated, and in

tandem the nature of the conclusions. The main conclusion seems to be that autoconversion is well prescribed, yet the radiative impacts of different schemes differ. It would be a great benefit to include the radiative impact of the clouds below the Mauritsen limit.
* * *
[Figure]

---

## Referee Comment (RC3) · Anonymous Referee #3 · 22 Jul 2019

In this paper, the authors investigate whether the linear relationship between LWC and CDNC observed in clouds during the NETCARE field campaign can be partially explained as due to autoconversion. They find that autoconversion is an important contributor to this relationship, although some of the variability is captured by their model even in the absence of autconversion, and therefore must be due to other sources. The paper is generally clearly written, and I feel that it warrants publication, provided that the following comments are addressed.

General comments:

1. Is cloud droplet sedimentation (i.e. gravitational settling) included in SCM-ABLC? This should be explicitly stated, especially for the interpretation of the "no rain" case results. If not, then it is possible that this process would allow the model to better simulate the cases with the lowest CDNC values, due to the larger modelled cloud droplet sizes, regardless of which autoconversion scheme was used. For the case with a CDNC of 5 cm-3 in particular, the large modelled size of the cloud droplets could allow cloud droplet sedimentation to be significant, even in the absence of collision-coalescence processes to grow the cloud droplets to drizzle drop sizes. This should be discussed.

2. Does the cloud vertical extent vary between simulations with different autoconversion schemes? Does the relationship between cloud vertical extent and CDNC differ between autoconversion schemes? The sensitivity test described on page 22 suggests that this could be important for shortwave radiative fluxes. This should therefore also be included in the discussion of aerosol indirect radiative effects on page 22.

3. I do not see any justification shown for the authors' inconsistent use of either $p = 0.01$ and $p = 0.05$ as thresholds for significance. The use of different thresholds is most jarring in the abstract, on P21, lines 535-539, and on P21-22, lines 541-546. In all three locations, a value of $p > 0.01$ is used to imply that no significant difference exists, and $p < 0.05$ is used to imply that a significant difference does exist.

I would suggest that if the authors have a justification for using a particular p value as the threshold for significance for this set of data, that it be included. If multiple different p value thresholds are used, this should be justified.

Alternatively, the discussion of p-values could be rephrased such that statistical significance is not a binary value: p-values and significance would thus be treated similarly to the way that $R^2$ values and correlation are currently discussed in this and many other manuscripts.

Specific comments:

P2, lines 48-52: Changes in CDNC have also been linked to changes in cloud-top radiative cooling, which subsequently affects LWP through changes in cloud vertical thickness (e.g. Possner et al., 2017).

P4, lines 98-101: At least for the context of the observations, please offer a numerical value for the Mauritsen limit here. Also, the definition of the Mauritsen limit as "it is a proposed threshold for aerosol concentration, below which cloud droplets that form grow to sizes large enough to precipitate" is imprecise. Some cloud droplets will grow to sizes large enough to precipitate in many clouds with larger aerosol concentrations. And if it was true that below the Mauritsen limit all cloud droplets immediately grew to precipitation sizes, then no droplets would be observed in the cloud droplet size range. Please provide more precise definitions of the Mauritsen limit and the tenuous cloud regime. It might be helpful to define the tenuous cloud regime first, and to define the Mauritsen limit in that context.

P5, lines 146-147: For what reason is it expected that the number of larger droplets was negligible during these flights?

Section 2.1: It would be helpful to give a description of any available observations of precipitation (or the absence thereof).

P9, lines 241-243: This was not completely clear to me. Was the modification of the boundary-layer height an iterative process, requiring multiple simulations? Was the location of the LWC maximum in each timestep compared to the location of observed LWC maximum, and the boundary-layer height adjusted online during a single simulation?

P11, lines 310-315: If aerosols were omitted in the radiative transfer calculations, then why were their optical properties computed? Also, please directly reference the parameterizations used for the optical properties.

P13, line 370: This function cannot be correctly described as a linearization. It would

be better to call it a piecewise function.

P22, line 551: Considering the discussion of statistical significance that precedes it, a different word than "significantly" should be used here.

Figures 2, 3: Would it be possible to have a single label for the corresponding lines and points in the legends? For example, could "Observations" be listed only once, with both the green line and green triangle? This would make the legends much clearer.

Technical corrections:

P1, line 21: "clouds,." -> "clouds."

P3, line 70: "consider the compare" please rephrase.

P9, line 246: please replace the dash after "SCM-ABLC" with a colon.

P9, line 257: please add a space after "ocean".

P9, line 257: please add "implemented" after "As".

P20, line 522: Perhaps it would be clearer to say "radiative transfer calculations" instead of "model runs". The current phrasing leaves some ambiguity between the SCM-ABLC simulations (which are based on observations) and the radiative transfer calculations based on in-flight observations only.

References:

Possner, A., Ekman, A. M. L., and Lohmann, U.: Cloud response and feedback processes in stratiform mixed-phase clouds perturbed by ship exhaust, Geophys. Res. Lett., 44, 1964–1972, https://doi.org/10.1002/2016GL071358, 2017.
* * *

---

## Author Comment (AC1) · 29 Sep 2019

For clarity, reviewer comments are shown in normal font, response to reviewer comments are highlighted in yellow and **text edited** in the revised manuscript is shown in bold, underlined and highlighted.

**Reviewer 1:**

**1 Comments on presentation**
I am not an expert on arctic clouds, but globally, the question of LWP adjustments to Nd changes is extremely important in the context of rapid adjustments (formerly the "cloud lifetime effect") to the radiative forcing by aerosol–cloud interactions (formerly the "Twomey effect"). See, e.g., Gryspeerdt et al. (2019 ACP); Rosenfeld et al. (2019 Science); Mulmenstadt and Feingold (2018 Current Climate Change Reports). It might help to make the connection to this literature in the introduction. As currently written, the introduction left me confused whether this is an aerosol–cloud paper (as the focus on Nd would suggest); or a feedbacks paper (as the statement about rapid warming would suggest); or a paper on the interaction between the two, in which case, perhaps cite Nazarenko et al. (2017 JGR) or Lohmann (2017 JGR) or an arctic equivalent, if that exists. However, from the main text, I think it's an ACI paper, so I would focus the introduction on LWP adjustments to the Twomey effect.

We would like to thank the reviewer for pointing out the confusion. We agree that connections to the literature should be clearer given the complex nature of cloud microphysical processes and their role for climate. As explained in more detail in the revised version of the manuscript, impacts of cloud microphysical processes on temperature trends are typically different for Arctic clouds than for clouds at lower latitudes. In particular, the Mauritsen limit is a proposed threshold for the aerosol concentration, where changes in the CCN (hence CDNC) results in net warming due to longwave effects, as referred to by Leaitch et al. (2016).

The focus of our paper is indeed aerosol-cloud interactions. Although the majority of the papers suggested by the reviewer focus on tropical and/or mid-latitude convection-based clouds, we have nevertheless added citations to the work of Mulmenstadt and Feingold (2018), Gryspeerdt et al. (2019), and Rosenfeld et al. (2019) in the introduction. We have also added more information about the cloud regime in question to the introduction, in hopes that it will help clarify the reviewer's confusion. The text now includes the following passages in the introduction:

Paragraph 1:
    **"Microphysical properties of Arctic clouds are sensitive to changes in cloud condensation nuclei (CCN) concentrations (Coopman et al., 2018) as is cloud radiative effect (Rosenfeld et al., 2019)."**

Paragraph 1:
    **"The present investigation involving the relationship between LWC and CDNC has also been found to vary geographically in other regions of the world (Gryspeerdt et al., 2019)."**

Paragraph 2:
    **"Others have pointed out the inherent difficulty of reconciling the abstraction of autoconversion from the physical processes in the cloud as well (Mülmenstädt and Feingold, 2018)."**

and paragraph 3:

**"Recent observations by Leaitch et al. (2016) showed a strong linear relationship between LWC and CDNC in low altitude, relatively horizontally homogeneous, liquid clouds in the summertime Canadian Arctic with weak influences by outside mixing processes aside from the top and bottom of the clouds. The clouds were formed as air advected over cold water, rather than by lifting, and as such differ significantly from the adiabatic cloud concept model. In these clouds, LWC was approximately constant from the top of the cloud to the bottom of the observations, implying that the cloud did not form by lifting and condensation (Leaitch et al., 2016). The clouds were also persistent in time with no evidence of significant precipitation, hence likely in a quasi-equilibrium state."**

We believe that these changes more clearly illustrate the relevance of the study to a broader discussion of the role of aerosol/cloud interactions in the climate system. More specifically, the goal of this paper is to contribute to this complex discussion by contributing to the knowledge of the nature of autoconversion in Arctic clouds during the NETCARE campaign.

Citations were also added for these papers.

The other major presentation question I had after reading the paper was what justified this focus on pure liquid clouds. Perhaps this betrays my ignorance, but I thought the most radiatively important low cloud type was mixed-phase in the Arctic, even in summer (Shupe and Intrieri, 2004?; de Boer et al., 2009).

Shupe and Intrieri (2004) found that "Overall, low-level stratiform liquid and mixed-phase clouds are found to be the most important contributors to the Arctic surface radiation balance," so liquid water clouds are relevant in this environment. Additionally, many Arctic cloud papers are based around Utqiaġvik (formerly Barrow), Alaska, during the Arctic haze period, where mixed phase clouds are present, while our study is based near Resolute Bay, Nunavut, during July, where liquid clouds were prevalent.

**2 Comments on the analysis**

My main substantive concern is that I am not convinced the findings are statistically robust. If I understand the analysis correctly, the authors simulated 11 cloud profiles with different mean Nd and LWC. In the observations, cloud-mean LWC is proportional to cloud-mean Nd, based on these 11 data points. The authors then tried different micro-physical schemes to determine which one is best able to reproduce the observations. A setup without autoconversion gives the worst regression coefficient, and based on this, the authors conclude that autoconversion is responsible for the proportionality.

My concerns, in detail, are:

1. The no-autoconversion setup does pretty well, actually. It simulates higher LWC than the other setups, but that is to be expected, because a big sink process for LWC is turned off. The slope in Fig. 2 looks indistinguishable from the setups that are claimed to work better.

We have added the 95% confidence interval for the slope to Table 2: the "no rain" case has a larger variance, a lower $R^2$ value, and a larger interval than the other cases, suggesting that the appearance of the slope can be misleading on its own. We have also corrected the variance of the LWC column in Table 2, which was previously average variance. Nevertheless, we agree that there is relatively good

agreement of model results with a few of the observations at CDNC > approx. 60 cm$^{-3}$ for the no-autoconversion setup, which provides evidence for the importance of the meteorological situation on the observed relationships beyond impacts of cloud microphysical processes on the LWC/CDNC relationship, as mentioned in the text. The model provides a tool for the quantification of these different influences on the LWC/CDNC relationship (Table 2).

2. The constant Nd runs also have pretty large slopes, which would indicate to me that a big part of the LWC increase is not due to autoconversion, or at least not due to the parameterized Nd dependence in the autoconversion rate.

The constant CDNC runs have low R$^2$ values and the "All CDNC 112/cm$^3$" case also has a large confidence interval, although the "All CDNC 5/cm$^3$" case does better by these measures than some of the schemes from the literature. We acknowledge that there are other processes affecting the LWC, and we do not expect the autoconversion to be the sole driver of the relationship. This is now emphasized in section 3.1 where we say:

"**Interestingly, simulations with the simplified parameterizations that do not account for effects of CDNC on autoconversion ('All CDNC 5/cm$^3$' and 'All CDNC 112/cm$^3$') also produce LWC values that are similar to the observed values for some of the flights, but have lower R$^2$ values compared to results with L&D and K&K parameterizations (see Table 2). However, the relative size of the 95% confidence interval from the 'All CDNC 112/cm$^3$' is large in comparison to the autoconversion cases and the observations (Table 2)."**

3. Continuing on that thought: in an attribution of an observed relationship to a candidate process, I would want to see some discussion on why other candidate processes are eliminated. Eliminating candidate processes is, of course, what models excel at, but confirming them, not so much (Oreskes et al. 1994, maybe?). First, I would want to know whether the clouds are adiabatic. Then, I would wantto know what stage in their life cycle they are in. At that point, a clearer picture may start to emerge; for example, in an adiabatic cloud, the vertically averagedLWC increases with cloud geometric thickness (thermodynamic conditions being equal), and the geometric thickness increases withNd(Pincus and Baker, 1994),purely from energy budget considerations.

Based on those concerns, I think a more convincing way to approach the problem would be first to build a conceptual model of the clouds and then to eliminate candidate processes (which probably requires numerical modeling), rather than to pick one process seemingly arbitrarily and trying to "confirm" it (because, as we know, science is the process of hypothesis refutation, not hypothesis confirmation).

The reviewer seems to be approaching the concept of the cloud as a typical updraft scenario with some vertical mixing due to cloud top cooling. However, although some of the higher-altitude clouds from the NETCARE campaign were formed from the more common lifting process, those discussed here were all low-altitude clouds with properties suggesting more similarities to advection fog. The introduction has been edited to clarify this point, and now states:

"**Recent observations by Leaitch et al. (2016) showed a strong linear relationship between LWC and CDNC in low altitude, relatively horizontally homogeneous, liquid clouds in the summertime Canadian Arctic with weak influences by outside mixing processes aside from the top and bottom of the clouds. The clouds were formed as air advected over cold water, rather than by lifting, and as such differ significantly from the adiabatic cloud concept model. In these**

**clouds, LWC was roughly constant from the top of the cloud to the bottom of the observations, implying that the cloud did not form by lifting and condensation (Leaitch et al., 2016). The clouds were also persistent in time with no evidence of significant precipitation, hence likely in a quasi-equilibrium state."**

Overall, we believe that key physical processes to the formation of the clouds are sufficiently accounted for in the simulations, as is also evident from the good agreement of model results with observations. Although the model is conceptually relatively simple, e.g. due to the omission of 3D transport processes, it is not obvious whether even simpler modelling frameworks exist that may also help to explain the observed relationships between LWC and CDNC.

My methodological comment is on letting the single-column model run to equilibrium. Actual clouds do not reach equilibrium, because precipitation acts as a condensate sink that (along with evaporation) causes the clouds to dissipate. In your method, the condensate loss by precipitation is balanced by moisture supply by advection, allowing the cloud to live forever. Clouds that live forever seem like a major limitation in a study on the cloud lifetime effect.

On the other hand, the model obviously needs to spin up.

This is a problem that the authors need to solve, but two suggestions they may find useful are:

1. Argue that clouds at any given point in time are in "quasi"-equilibrium. This is actually an assumption in many GCM parameterizations, i.e., the state the GCM tries to capture is representative of a cloud field averaged over a fairly long time step (30 minutes). However, I don't know if I would buy the argument for an individual cloud.

While the model is run to equilibrium, and the cloud at any point in time is not exactly at equilibrium, the clouds in question were highly persistent, indicating that the clouds were likely in quasi-equilibrium. We have noted the likely quasi-equilibrium state on lines 88-89, which state:

**"The clouds were also persistent in time, hence likely in a quasi-equilibrium state."**

Although we clearly do not address aerosol impacts on clouds or the cloud lifetime effect in our study, it should be emphasized that the model predicts the LWC and its response to autoconversion, which is a fundamental aspect of the Twomey effect. We also would like to clarify that the moisture budget in the simulation is a near-balance between surface fluxes, precipitation, and turbulent transport of moisture at the top of the cloud layer. For the shallow clouds that were observed, time scales of these processes are likely short relative to the time scale of advective transport into the cloud layer. The observations indicate that the clouds and atmospheric flow were fairly uniform over many tens of kilometres, so these assumptions seem justified. As such, it is not obvious to us that a fully prognostic 3D simulation of the observed cloud deck would necessarily produce much more realistic cloud vertical LWC profiles than a meteorologically highly constrained single column model.

2. Spin up the model with one Nd, then observe the transient behavior when you abruptly change to a different Nd. There is a series of papers by Andrew Gettelman (2015) on SCM studies of different cloud microphysics that might provide insight.

This is a very interesting suggestion but would require substantive work to do it justice and could be the focus of future work. We have added this in the conclusion, which now states:

**"It may also be of interest to compare these findings to a large-eddy simulation model. Another interesting future direction would be to probe our assumption that the cloud is in equilibrium. This could be accomplished by changing the CDNC abruptly after the model spin-up to observe the transient behaviour of the model microphysics, as performed by Gettelman (2015)."**

**Reviewer 2:**

This paper uses previously published observations (Leaitch ACP2016) of Arctic boundary layers of cloud liquid water content and droplet number to study the response of three autoconversion schemes and then considers the radiative properties of one cloud.

New data analysis demonstrates the linear relationship between cloud drop number concentration and liquid water path for these clouds which is a useful addition to the observational record. A number of samples in cloud were of low droplet number, in theCCN limited regime, referred to as the Mauritsen limit. Only very few samples were collected in this regime but the linear relationship appears to hold. As mentioned in the text, there is significant variability within the data, perhaps related to background meteorological conditions. It is therefore difficult to ascertain the significance of the result more broadly.

A main focus of the analysis and result is the comparison of three autoconversion schemes with the aim of investigating the impact of autoconversion on the LWP:CDNC relationship. All three of the autoconversion schemes appeared to perform well, suggesting that the process is in fact well constrained, even for the Arctic. The authors then demonstrate that a combination of two schemes, one for the CCN limited regime, and one for larger concentrations of CCN can improve on the performance of a single scheme across the full phase space. Again, the limited data makes this an intriguing but not completely satisfying result, with no attempt to explain, other than to invoke other processes such as turbulence and mixing.

Once the autoconversion schemes have been considered there is a section that investigates the radiative properties of clouds. The paper seems to lead towards a comparison of the radiative properties of clouds that are and are not CCN limited. However, only a cloud that is above this CCN limit is investigate, to the detriment of the work. Further, following the comparison of autoconversion schemes, these are shown to not have a large impact on the radiative properties of the clouds. The finding that the modelled radiation using the autoconversion schemes is different from observations in theJuly 8th cloud warrants further investigation and may be a useful result. Having only one such case though, is not likely to be sufficient to inform the modelling community of changes that might need to be made to the representation of aerosol indirect effects.

Numerous tables give details of the linear fit parameters, which whilst required, are not so easy to interpret. I would suggest that some measure of the uncertainty / significance is added to the plots to allow the reader to make an informed assessment. This would be useful on Figures 3, 4 and 5. It may also assist the reader to combine those panels in to a single figure. The aims of the paper should be more clearly stated, and in tandem the nature of the conclusions. The main conclusion seems to be that autoconversion is well prescribed, yet the radiative impacts of different schemes differ. It would be a great benefit to include the radiative impact of the clouds below the Mauritsen limit.

We would like to thank the reviewer for taking the time to provide their thoughtful comments. The reviewer's main concern seems to stem from the limited number of cases that were simulated, especially below the Mauritsen limit. We share these concerns, but are unfortunately limited by the data available from the study. It should be emphasized that this paper is not intended to serve as an overview of all that is possible in Arctic clouds. Instead, we can only examine a few low, liquid clouds that were observed to be persistent in the summer months when the Arctic is generally regarded to be quite pristine. We do extrapolate some larger conclusions from this small dataset about what is possible, but they are not intended as a limitation, but more of a broadening of possibilities and indication of avenues

of future interests to see whether our results can be applied to other datasets or if they are more unique to the location. To emphasize the limitations of our study, we have reworded some of the text in the conclusion to read as follows:

**"It is important to note that our observations below the Maurtisen limit only consisted of 3 profiles and that our conclusions are dependent on this limited data set. It would be of interest to examine whether this regime change can be reproduced with more data, in other parts of the summer Arctic, and with other models."**

Turbulence and mixing are complex processes with effects that are broad and difficult to quantify. We believe it is reasonable to assume that these processes are acting in the clouds in ways that are not well-represented by our models.

We did not model the radiative impacts of the clouds below the Mauritsen limit due to the limited number of vertical profiles in these clouds. Not only were there only three flight profiles into the clouds on July 5 and 7 in total, but also the solar zenith angle and surface albedo varied between these two flights as well as with July 8. Overall, this resulted in uncertainties that were too large to allow useful comparisons between all the profiles. This has now been clarified in Section 2.3; the text now reads:

**"Only profiles from July 8 are used for the radiative transfer calculations. The flights from July 5 and 7 were not analyzed due to the different solar zenith angles, the different surface albedos, the small number of available cloud profiles, and the possible effects of a different regime at lower CDNC. This resulted in uncertainties that would have made meaningful comparisons difficult."**

To address the reviewer's concerns about the figures, we have now grouped them together for easier reference, while preserving the higher quality images. We have also added the 95% confidence interval for the slope to tables 2 and 3, which will aid in the interpretation of the linear fit parameters. However, we believe that adding uncertainty or significance directly to the plots would add more visual confusion rather than lessen it and have not included this in our new figures.

**Reviewer 3:**

1. Is cloud droplet sedimentation (i.e. gravitational settling) included in SCM-ABLC? This should be explicitly stated, especially for the interpretation of the "no rain" case results. If not, then it is possible that this process would allow the model to better simulate the cases with the lowest CDNC values, due to the larger modelled cloud droplet sizes, regardless of which autoconversion scheme was used. For the case with a CDNC of 5 cm-3 in particular, the large modelled size of the cloud droplets could allow cloud droplet sedimentation to be significant, even in the absence of collision-coalescence processes to grow the cloud droplets to drizzle drop sizes. This should be discussed.

The reviewer brings up an important point. The cloud scheme in the SCM-ABLC does not include gravitational settling of cloud droplets. We have added the following sentence on lines 453-455:

**"Due to the lack of droplet sedimentation in the model, the droplets in the 'All CDNC 5/cm$^3$' case are likely to be very large, possibly resulting in more autoconversion than expected and lower LWC values in this simulation."**

2. Does the cloud vertical extent vary between simulations with different autoconversion schemes? Does the relationship between cloud vertical extent and CDNC differ between autoconversion schemes? The sensitivity test described on page 22 suggests that this could be important for shortwave radiative fluxes. This should therefore also be included in the discussion of aerosol indirect radiative effects on page 22.

Most of this process was originally described in section 2.3.2 in relation to the cloud profiles that were used for the radiative transfer model, but also applies to the discussion for Figure 2. We have added the following paragraph as section 2.2.6, and edited section 2.3.2 as not to be overly repetitious.

**"The cloud vertical extent produced by the SCM-ABLC can differ slightly between different autoconversion schemes, and does in some simulations. However, since the aircraft observations used in our comparisons do not include the entire cloud but only the uppermost part of it, we have focussed on comparing the thicknesses equivalent to the observed portion of the clouds rather than examining the modelled vertical extent. For each observed profile, we used the thickness of cloud measured down from the modelled cloud top to the penetration depth of the aircraft into the cloud during the NETCARE flights. Parts of cloud below the lowest flight level of the aircraft were omitted to avoid only relying on model output."**

3. I do not see any justification shown for the authors' inconsistent use of either p=0.01and p=0.05 as thresholds for significance. The use of different thresholds is most jarring in the abstract, on P21, lines 535-539, and on P21-22, lines 541-546. In all three locations, a value of p>0.01 is used to imply that no significant difference exists, and p<0.05 is used to imply that a significant difference does exist. I would suggest that if the authors have a justification for using a particular p value as the threshold for significance for this set of data, that it be included. If multiple different p value thresholds are used, this should be justified. Alternatively, the discussion of p-values could be rephrased such that statistical significance is not a binary value: p-values and significance would thus be treated similarly to the way that R^2 values and correlation are currently discussed in this and many other manuscripts.

We agree that it is confusing and we have elected to use p=0.05 as the threshold everywhere.

P2, lines 48-52: Changes in CDNC have also been linked to changes in cloud-top radiative cooling, which subsequently affects LWP through changes in cloud vertical thickness (e.g. Possner et al., 2017).

We have added a sentence to reflect this as well, as follows:
**"Depending on the amount of moisture in the free troposphere, changes in the CDNC may also positively or negatively affect the LWP via increased cloud top radiative cooling enhancing turbulent mixing and hence entrainment near the top of the cloud (Possner et al., 2017; Chen et al., 2015; Ackerman et al., 2004), or via precipitation."**

Citations were also added for these papers.

P4, lines 98-101: At least for the context of the observations, please offer a numerical value for the Mauritsen limit here. Also, the definition of the Mauritsen limit as "it is a proposed threshold for aerosol concentration, below which cloud droplets that form grow to sizes large enough to precipitate" is imprecise. Some cloud droplets will grow to sizes large enough to precipitate in many clouds with larger aerosol concentrations. And if it was true that below the Mauritsen limit all cloud droplets immediately grew to precipitation sizes, then no droplets would be observed in the cloud droplet size range. Please provide more precise definitions of the Mauritsen limit and the tenuous cloud regime. It might be helpful to define the tenuous cloud regime first, and to define the Mauritsen limit in that context.

We have added two previously-computed numerical values, as well as an explanation of the tenuous cloud regime and clarification of the Mauritsen limit on lines 99-108. The text now reads:
**"Mauritsen et al. (2011) proposed that a tenuous cloud regime exists when cloud formation is limited by the available CCN, wherein the low CDNC causes rapid growth from vapour deposition resulting in droplets large enough to fall. This was expanded by Leaitch et al. (2016), who introduced the Mauritsen limit as a threshold for the aerosol concentration, below which an increase in the CCN (hence CDNC) results in net warming due to longwave effects. As such, clouds with aerosol concentrations below the Mauritsen limit are presumed to be in the tenuous-cloud regime. Previously determined numerical values of the Mauritsen limit have included 10 cm$^{-3}$ (Mauritsen et al., 2011) and 16 cm$^{-3}$ (Leaitch et al., 2016), but the concept is not tied to specific droplet number concentrations as the environment can affect the threshold (Leaitch et al., 2016; Mauritsen et al., 2011)."**

P5, lines 146-147: For what reason is it expected that the number of larger droplets was negligible during these flights?

We concluded this based on the statistics presented by Leaitch et al. (2016) which reported that the 95th percentile of the volume mean diameter measured by the FSSP during low level clouds was 31 µm, which was far below the detection limit of 45 µm. The sentence in the paper now reads as follows:
**"However, we expect that the number of larger droplets was negligible in this work, as the 95th percentile volume mean diameter observed in low altitude clouds by Leaitch et al. (2016) was 31 µm, far below the upper size limit."**

Section 2.1: It would be helpful to give a description of any available observations of precipitation (or the absence thereof).

A PMS 2D-C greyscale probe present on the aircraft found no ice crystals or water droplets with diameter greater than 100 μm during the flights modelled in our study (Leaitch et al., 2016). We have included the following in the text in the paragraph before Section 2.1.1:

**"As per Leaitch et al. (2016), no ice crystals or water droplets with diameter greater than 100 μm were detected by the PMS 2D-C greyscale probe in any of these clouds, suggesting that these clouds were not precipitating. However, the low altitude clouds with very low droplet concentrations on July 5 and 7 had some droplets large enough in size (greater than 30 μm) that their settling speed was high enough to possibly be viewed as precipitation."**

P9, lines 241-243: This was not completely clear to me. Was the modification of the boundary-layer height an iterative process, requiring multiple simulations? Was the location of the LWC maximum in each time step compared to the location of observed LWC maximum, and the boundary-layer height adjusted online during a single simulation?

We agree that this was not clear in the original text. The section now reads:

**"For each case, the boundary layer height was estimated from the height of the base of the observed temperature inversion. The SCM-ABLC was then run for estimated modelled boundary layer heights within 30 m of that height. The height that resulted in a LWC profile most qualitatively similar to the observed was then used for all subsequent simulations for that case. The model LWC profile was averaged over the final 50 hours of the simulation and then used for all later runs for that case; the procedure was repeated for all cases."**

P11, lines 310-315: If aerosols were omitted in the radiative transfer calculations, then why were their optical properties computed? Also, please directly reference the parameterizations used for the optical properties.

We have removed the reference to aerosol calculations as we had indeed not calculated their optical properties. We have also added the references for the liquid water cloud optical property parameterizations. The text now reads:

**"Absorption by gases is computed using the correlated-$k$ method (von Salzen et al., 2013; Lacis and Oinas, 1991). The optical properties of liquid clouds are computed using the parameterizations referenced by von Salzen et al. (2013), separately for solar (Dobbie, Li, and Chýlek, 1999) and infrared (Lindner and Li, 2000) wave numbers."**

and

"**Aerosols were omitted in the radiative transfer calculations due to their relatively small effects on the radiative fluxes compared to those due to the clouds."**

P13, line 370: This function cannot be correctly described as a linearization. It would be better to call it a piecewise function.

We thank the reviewer for pointing this out. This has been modified to now read:

**"Based on these results, we constructed a piece-wise function based on the two linearizations of the closest-fitting results to observations, called "L&D and K&K" corresponding to the combination of L&D and K&K schemes, with the K&K scheme at CDNC < 20/cm$^3$ and the L&D scheme at higher CDNC."**

P22, line 551: Considering the discussion of statistical significance that precedes it, a different word than "significantly" should be used here.

This has now been rephrased to read:
    **"This may require further investigation as the L&D and Wood schemes differ only by a constant, while the K&K scheme uses an additional variable as well as different constants than those two."**

Figures 2, 3: Would it be possible to have a single label for the corresponding lines and points in the legends? For example, could "Observations" be listed only once, with both the green line and green triangle? This would make the legends much clearer.

This has been done for all figures.

Technical corrections:
P1, line 21: "clouds,." -> "clouds."

Done.

P3, line 70: "consider the compare" please rephrase.

Done.

P9, line 246: please replace the dash after "SCM-ABLC" with a colon.

Done.

P9, line 257: please add a space after "ocean".

Done.

P9, line 257: please add "implemented" after "As".

Done.

P20, line 522: Perhaps it would be clearer to say "radiative transfer calculations" instead of "model runs". The current phrasing leaves some ambiguity between the SCM-ABLC simulations (which are based on observations) and the radiative transfer calculations based on in-flight observations only.

Done.

**Additional Edits:**

The following modifications were made throughout the paper to improve the clarity and language:

**"Longwave/shortwave fluxes"** referring to the difference in those fluxes due to the cloud radiative effect throughout the paper have been better specified as **"longwave/shortwave cloud radiative effect"** or **"longwave/shortwave CRE."**

The current affiliation for Rashed Mahmood is now as follows (line 14): **\*- Now at Barcelona Supercomputing Center, Barcelona, Spain**

Line 20-21: NETCARE -> **Network on Climate and Aerosols: Addressing Key Uncertainties in Remote Canadian Environments**

Line 23 now reads: **"Of the three autoconversion schemes we examined, the scheme..."**

Lines 31-34 now read: **"In contrast, the downward longwave and shortwave cloud radiative effect at the surface for Wood and K&K schemes do not differ significantly (p=0.05) from the observation-based radiative calculations, while the L&D scheme differs significantly from the observation-based calculation for the downward shortwave but not the downward longwave fluxes."**

Line 41 now reads: **"As observed at other latitudes, for comparable liquid water content (LWC), ..."**

Lines 43-46 now read: **"However, the net radiative effect of cloud droplet size and number concentration can vary in sign in the Arctic due to the interplay between longwave and shortwave radiative effects when there are high surface albedo and large solar zenith angle (Curry et al., 1996)."**

Line 47: **"dominated"** -> **"controlled"**

Line 47-48: **"liquid water content"** -> **"liquid water path (LWP)"**

Line 49: removed **"Similarly,"**

Lines 50-51 now read: **"Model simulations without shortwave radiation have been used to show that it can..."**

Line 56: added **"e.g."** before the citation of Rosenfeld et al. (2014)

Line 73: added **"(three large eddy simulations and three numerical weather prediction models)"** to describe the models from the cited study.

Lines 79-82 now read: **"However, the study did not test different autoconversion parameterizations using the same model. Nor did the study compare the results of Arctic clouds with different CCN concentrations or rain formation schemes in the models (Stevens et al., 2018)."**

Line 106: **"keeping the CDNC"** -> **"the CDNC remains"**

Line 117-118: **"NETCARE"** -> **"Network on Climate and Aerosols: Addressing Key Uncertainties in Remote Canadian Environments (NETCARE)"**

Line 118-119: **"cloud droplet number concentrations"** -> **"CDNC"**

Lines 133 & 136: removed **"will"**

Lines 138-139: **"relying on"** -> **"as it uses"**

Line 141: **"resulting"** -> **"that might result"**

Line 143: **"will be"** -> **"that are"**

Lines 144-146 now read: **"Changes in the radiative balance of the simulated clouds due to differences from the autoconversion schemes are examined using an offline version of the radiative transfer model in CanAM4.3"**

Lines 151-152: **"Network on Climate and Aerosols: Addressing Key Uncertainties in Remote Canadian Environments (NETCARE)"** -> **"NETCARE"**

Lines 175-177 now read: **"(approximately 45 μm; Leaitch et al., 2016)"**

Line 186: **"2.2.1"** -> **"2.1.1"**

Lines 241-242: added **"Model results from the last 200 time steps, or 50 hours, were then averaged."**

Lines 252-253: **"cloud droplet number concentration"** -> **"CDNC"**

Line 290: replaced "-" with a colon

Line 301: added **"implemented"**

Lines 310-312 now read: **"The different representation of the autoconversion process in the L&D scheme results in stronger dependencies on LWC and CDNC"**

Line 320: added **"empirically-calculated"**

Lines 327-330 now read: **"All were originally developed for the mid-latitudes so as part of our study, we will be evaluating their performance in summer Arctic low clouds."**

Line 358: **"most important"** -> **"main"**

Lines 376-377: **"liquid water path (LWP)"** -> **"LWP"**

Lines 378-380: removed description and added **"as described in Section 2.2.6"**

Line 380: added **"again"**

Line 384: **"Section 3.3" -> "Section 3.2"**

Line 392: **"inputting" -> "setting"**

Line 400: added **"spanning"**

Line 418: added **"3.1 SCM-ABLC"** as a section header

Line 420: **"expected from" -> "observed by"**

Line 424: **"driver" -> "source"**

Line 443: added **"theoretical"**

Line 475: **"high observed" -> "greater"**

Lines 494-495: **"debatable" -> "likely dependent on the environment"**

Line 528: **"as the case with no autoconversion" -> "in this instance, since the no-autoconversion case"**

Lines 534-538: We removed references to studies about cumulous-type clouds, as they were inappropriate here.

Line 540: **"We also" -> "since we"**

Lines 541-543 now read: **"Future work may have to better incorporate subgrid-scale cloud mixing processes in models."**

Line 545 now reads: **"3.2. Radiative fluxes"**

Line 551: **"cloud inputs from observations" -> "observed cloud properties"**

Lines 586-588 now read: **"Table 3. Summary of linear fits of radiation model CRE. See main text for description of cases."**

Lines 594-597 now read: **"A similar decreasing linear relationship exists for the downward shortwave CRE at the surface (Figure 4). However, there is no significant difference at p=0.05 (see Table 4) in the downward shortwave CRE at the surface between each scheme and the observation-based radiative transfer calculations on July 8 except for the "L&D" and "All CDNC 5/cm$^3$" cases."**

Lines 599-602 have been removed.

Lines 623-624 now read: **"the K&K scheme uses an additional variable as well as different constants than"**

Lines 645-646 now read: **"the three autoconversion parameterizations used in this study."**

Lines 649-652 now read: **"In particular, the change in shortwave CRE is much greater for the K&K parameterization than calculation based on observations, which is consistent with the particularly strong non-linear dependency of this parameterization on CDNC."**

Line 666: **"of the world where CDNC are higher"** -> **"with greater CDNC"**

Line 675: added **"which may also be interesting to reexamine with a larger dataset"**

The following citations were also added:

Dobbie, J. S., Li, J., & Chýlek, P.: Two and four stream optical properties for water clouds and solar wavelengths. J. Geophys. Res., 104, 2067–2079, 1999.

Lindner, T. H., and Li, J.: Parameterization of the optical properties for water clouds in the infrared. J. Climate, 13, 1797–1805, 2000.